# DNA methyltransferase inhibition upregulates MHC-I to potentiate cytotoxic T lymphocyte responses in breast cancer

Na Luo[1,2], Mellissa J. Nixon[2], Paula I. Gonzalez-Ericsson [3,4], Violeta Sanchez[3,4], Susan R. Opalenik[2], Huili Li[5], Cynthia A. Zahnow[6], Michael L. Nickels[7], Fei Liu[7], Mohammed N. Tantawy[7], Melinda E. Sanders[3,4], H. Charles Manning[7,8] & Justin M. Balko[2,4,9]

Potentiating anti-tumor immunity by inducing tumor inflammation and T cell-mediated responses are a promising area of cancer therapy. Immunomodulatory agents that promote these effects function via a wide variety of mechanisms, including upregulation of antigen presentation pathways. Here, we show that major histocompatibility class-I (MHC-I) genes are methylated in human breast cancers, suppressing their expression. Treatment of breast cancer cell lines with a next-generation hypomethylating agent, guadecitabine, upregulates MHC-I expression in response to interferon-γ. In murine tumor models of breast cancer, guadecitabine upregulates MHC-I in tumor cells promoting recruitment of CD8+ T cells to the microenvironment. Finally, we show that MHC-I genes are upregulated in breast cancer patients treated with hypomethylating agents. Thus, the immunomodulatory effects of hypomethylating agents likely involve upregulation of class-I antigen presentation to potentiate CD8+ T cell responses. These strategies may be useful to potentiate anti-tumor immunity and responses to checkpoint inhibition in immune-refractory breast cancers.

---

[1] Department of Anatomy and Histology, School of Medicine, Nankai University, Tianjin 300071, China. [2] Department of Medicine, Vanderbilt University Medical Center, Nashville, TN 37232-2310, USA. [3] Department of Pathology, Microbiology, and Immunology, Vanderbilt University Medical Center, Nashville, TN 37232-2310, USA. [4] Breast Cancer Research Program, Vanderbilt University Medical Center, Nashville, TN 37232-2310, USA. [5] Pathology Department Penn State Health Milton S. Hershey Medical Center, Hershey, PA 17033, USA. [6] Department of Oncology, The Sidney Kimmel Comprehensive Cancer Center at Johns Hopkins, Baltimore, MD 21287, USA. [7] Department of Radiology and Radiologic Sciences, Vanderbilt Center for Molecular Probes, Vanderbilt University Medical Center, Nashville, TN 37232-2310, USA. [8] Department of Biomedical Engineering, Vanderbilt University Medical Center, Nashville, TN 37232-2310, USA. [9] Department of Cancer Biology, Vanderbilt University Medical Center, Nashville, TN 37232-2310, USA. Correspondence and requests for materials should be addressed to J.M.B. (email: justin.balko@vanderbilt.edu)

Although many tumor types have benefitted from immunotherapy, breast cancer remains a largely immune-refractory disease. Clinical trials with single-agent PD-1 or PD-L1 therapy have yielded clinical responses, although the fraction of patients responding has been underwhelming in early reports[1,2]. Thus, most clinical trials in breast cancer have now resorted to combination therapies. We and others have shown that inhibition of the Ras/MAPK pathway in breast cancer and other cancer types can upregulate both class I and class II major histocompatibility complexes (MHC-I, MHC-II, respectively) on tumor cells leading to enhanced anti-tumor immunity and potentiation of response to anti-PD-1/L1 therapies[3,4]. These findings have been substantiated in early reports in gastrointestinal malignancy[5]. Consistent with the ideology that suppressed MHC expression on tumor cells is associated with poor anti-tumor immunity, several studies have identified markers of antigen presentation as correlates of response to immunotherapies targeting the PD-1/L1 axis[6–8]. Furthermore, tumors that are de novo resistant to immunotherapy, or respond and ultimately progress, acquire mutations which suppress antigen presentation (such as *β2-microglobulin* loss), or the MHC-response to interferon-γ (IFNγ) stimulation[9,10]. Thus, MHC-mediated antigen presentation appears to be a significant modifier of anti-tumor immunity and response to PD-1/L1-targeted therapy.

Epigenetic modulation has been reported to create long-lasting effects on anti-tumor immunity, and a small number of NSCLC patients that progressed while in an epigenetic therapy trial anecdotally responded at relatively high rates to subsequent immunotherapy, despite not having substantial responses to the original epigenetic treatment[11–13]. There have been a number of proposed mechanisms for the augmentation of anti-tumor immunity and the increased subsequent benefit from immunotherapy after epigenetic treatment (reviewed in refs. [14,15]). Such mechanisms include activation of expression of endogenous retroviral sequences leading to viral mimicry, pattern-recognition receptor activation and innate immunity[16,17]. Epigenetic therapy has also been reported to increase antigen presentation pathways (e.g., MHCs) in tumors and tumor cells[11–13,16,17]. However, the utility of epigenetic modulation in modifying the specific microenvironment of breast cancer is relatively underexplored. This is particularly important in that breast cancers are largely immune 'cold' (i.e., lacking substantial inflammatory profiles and lymphocytic infiltration) with reduced tumor-associated neoantigens[18–20]. Thus, a better understanding of how epigenetic therapies, such as DNA methyltransferase inhibitors (DMTi) can promote anti-tumor immunity in breast cancer is needed.

In this study, we explored the effects of guadecitabine, a next-generation DMTi on MHC-I/II expression and gene promoter methylation in breast cancer cells. We found that guadecitabine potently upregulated MHC-I, particularly in response to type-II interferon stimulation, and promoted the expression of chemokines which drive T cell recruitment, which was also observed in mice treated with guadecitabine. While enhancement of response to interferon stimulation following guadecitabine treatment was dependent on basal increases in NFκB activity, basal upregulation of MHC-I genes appeared to be directly driven by promoter hypomethylation. DMTi-mediated effects on MHC-I gene expression were confirmed in human breast cancer patients who received epigenetic therapy, and suggest the potential for combination strategies of DMTi with immune checkpoint agents, such as PD-L1. We demonstrate this in principle using two murine orthotopic breast cancer models.

## Results

**MHC-I gene expression is negatively correlated with methylation**. Given these data, we asked whether MHC expression was likely to be regulated at the epigenetic level by exploring DNA methylation of MHC-I and MHC-II genes in The Cancer Genome Atlas (TCGA) breast cancer dataset[21]. We observed substantial inverse correlations between methylation levels of both class I (*HLA-A; HLA-B; HLA-C*) and class II (*HLA-DRA*) gene promoters and mRNA expression (Fig. 1a). Using *CD8A* mRNA as a surrogate for T cell infiltration, we also observed an inverse correlation between *HLA-A*, *HLA-B*, *HLA-C*, and *HLA-DRA* methylation and *CD8A* mRNA expression (Pearson's correlation of −0.33, −0.52, −0.37, and −0.38, respectively). Thus, MHC-I (and -II) expression is likely to be suppressed in many tumors at the epigenetic level, which correlates with reduced CD8+ T cell infiltrate. Several MHC-I genes were found to be upregulated by epigenetic therapy in patients with solid tumors previously[11].

**Inhibition of DNMT1 induces MHC-I expression**. To test this hypothesis, we evaluated breast cancer cell lines representing ER+ (MCF7), HER2+ (BT474), triple-negative (HCC1395), and claudin-low (BT549) subtypes of breast cancer for their response to a next-generation DMTi (guadecitabine; 7 day treatment) which targets *DNMT1*. In all cases, MHC-I expression at the cell surface was enhanced in a dose-dependent manner after subsequent stimulation with IFNγ (Fig. 1b). MHC-II induction was more heterogeneously affected by guadecitabine priming, with only MCF-7 cells demonstrating a dose-dependent effect. Doses utilized for this effect did not induce overt cytotoxicity, but did suppress proliferation (Fig. 1c), particularly in long-term assays.

**Inhibition of DNMT1 affects MHC-I promoter methylation**. Next we evaluated the effect of guadecitabine in a murine orthotopic model of mammary cancer (MMTV-Neu). Cultured MMTV-Neu cells had an $IC_{50}$ of 0.05 µg/mL when treated with guadecitabine, which was similar to the pharmacodynamic activity of the agent on global 5-methylcytosine (5-me-C) content (Fig. 2a–c). As with human breast cancer cell lines, priming of MMTV-Neu cells with guadecitabine enhanced both MHC-I and MHC-II expression after IFNγ stimulation in a dose-dependent manner (Fig. 2d). However, *PD-L1 (CD274)* induction (a known IFNγ-responsive gene) was not affected by guadecitabine, which is consistent with the reduced inverse correlation of *PD-L1* gene expression with *PD-L1* DNA promoter methylation shown in Fig. 1a. This is in contrast to hematologic malignancies, where treatment with hypomethylating agents promoted expression of PD-L1[22]. DNA methylation-specific PCR demonstrated that guadecitabine decreased MHC-I (H2-D1 promoter methylation (2/3 primers; Fig. 2e), and this corresponded to a dose-dependent increase in both basal *H-2D1* mRNA expression and IFNγ-inducible *H-2D1* mRNA expression (Fig. 2f). Thus, DMTi treatment demethylates the MHC-I promoter for *H2-D1* increasing basal promoter transcription and possibly priming for IFNγ-inducible gene expression.

**DNMT1 inhibition alters the immune microenvironment**. Given that MHC expression in tumor cells is frequently repressed, and therapies that upregulate MHC expression have been shown to elicit anti-tumor immunity, we performed orthotopic injection of MMTV-Neu tumor cells in syngeneic *FVB/n* mice. When tumors reached >100 mm³, mice received daily intraperitoneal injections of guadecitabine at 1.5 mg/kg or 3 mg/kg, or diluent control. Daily injections were performed for 3 sequential days, followed by a 4-day rest period. At day 7, mice were sacrificed for molecular analysis. We observed a substantial dose-

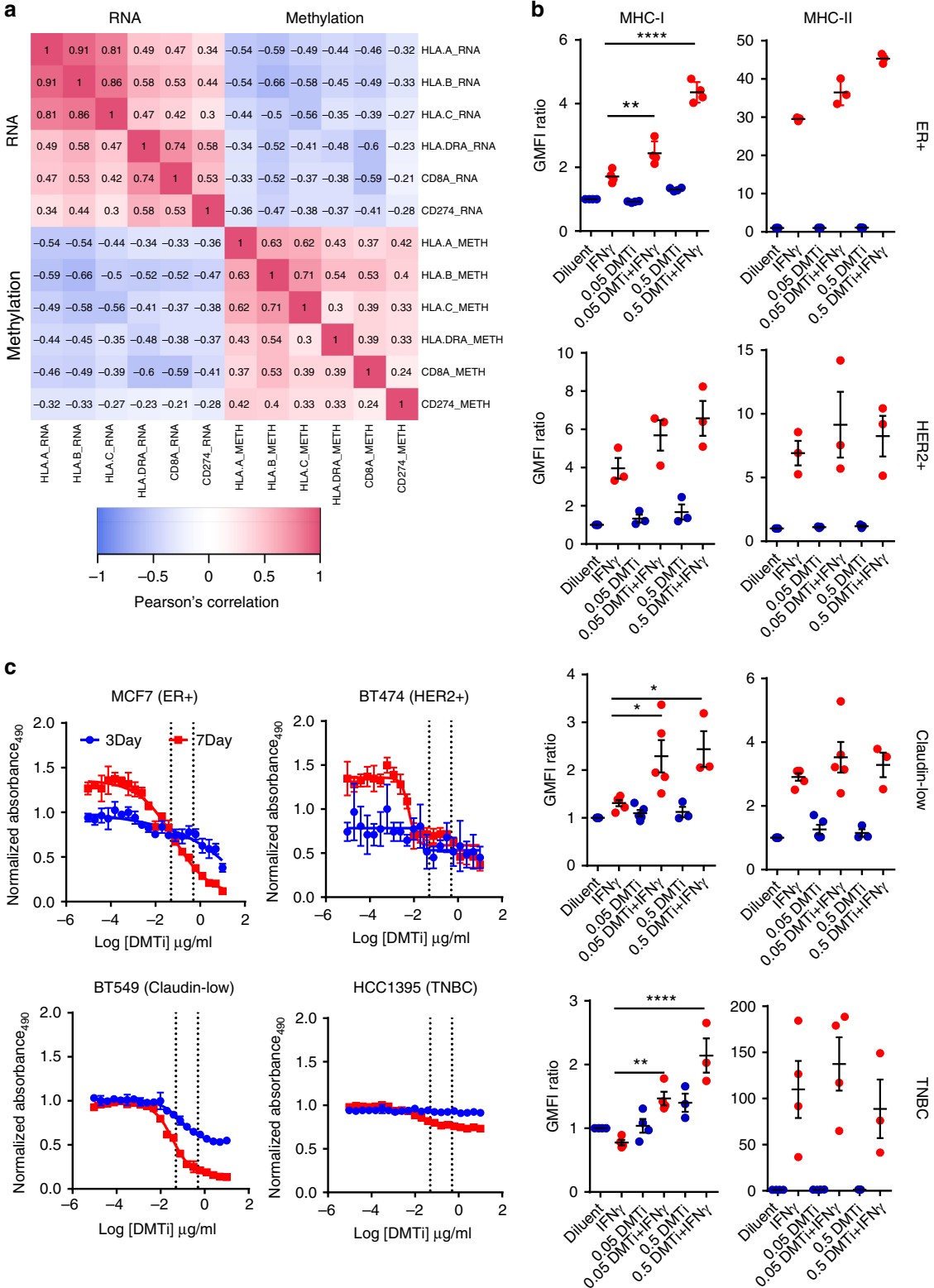

**Fig. 1** DMTi treatment augments MHC-I and MHC-II expression of human breast cancer samples. **a** Correlation matrix of methylation status and mRNA expression of MHC-I (HLA-A/B/C), MHC-II (HLA-DRA), and PD-L1 (CD274) from TCGA breast cancer patient data. Pearson's correlation coefficients are shown in box. **b** Geometric mean fluorescence intensity (GMFI) ratio of MHC-I and MHC-II in multiple subtypes of human breast cancer cell lines (ER +/MCF7; HER2+/BT474; Claudin-low/BT549; TNBC/HCC1395) under guadecitabine (DMTi) treatment + IFNγ stimulation. GMFI ratio of diluent group was used as the baseline. *$p < 0.05$; **$p < 0.01$; ****$p < 0.0001$ (one-way ANOVA with Tukey's multiple comparisons correction to compare individual groups). All data are means ± SEM. Each dot represents one independent experimental result. **c** Viability assay of multiple subtypes of human breast cancer cell lines (ER+/MCF7; HER2+/BT474; Claudin-low/BT549; TNBC/HCC1395) under DMTi treatment. All data are means ± SD

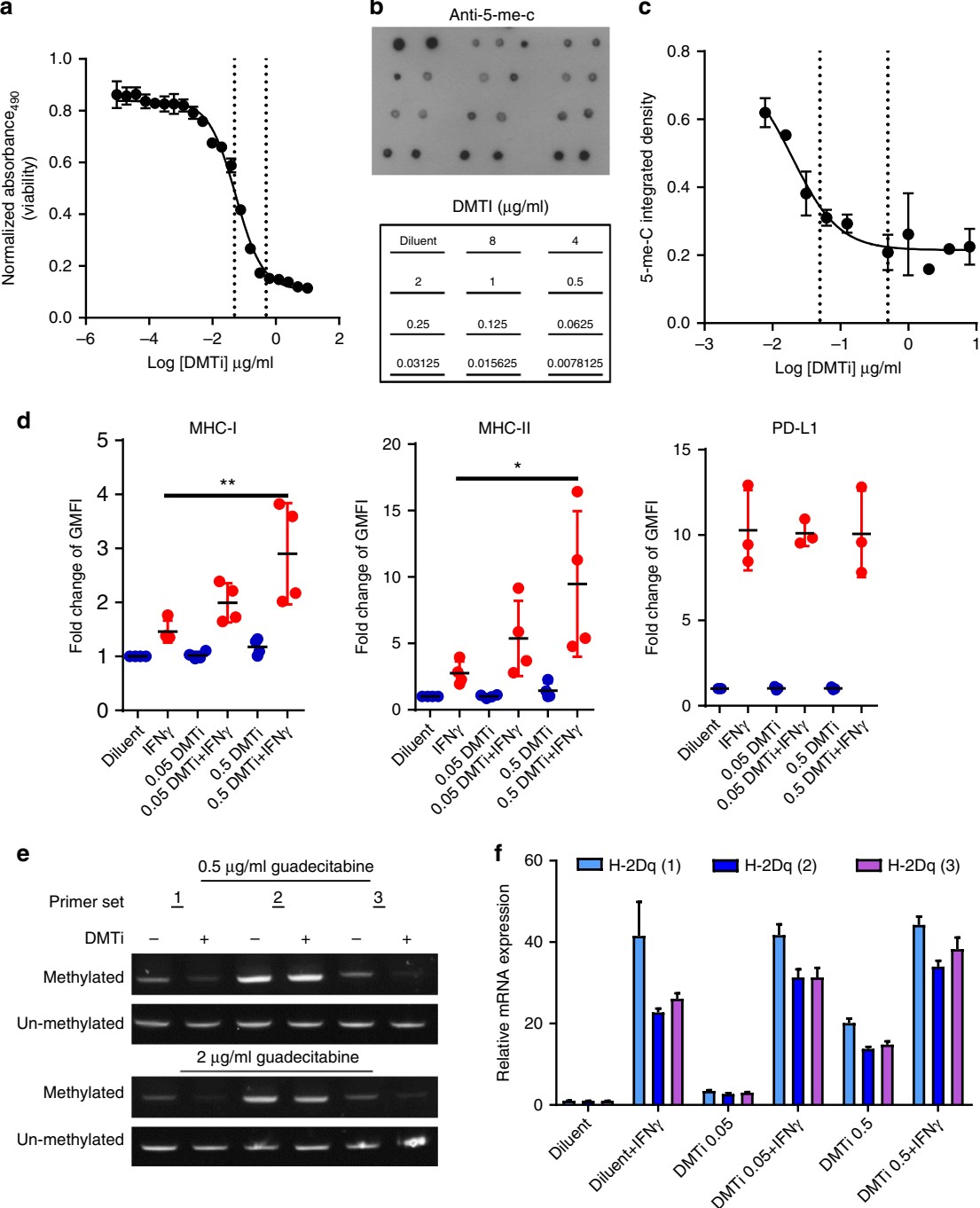

**Fig. 2** DMTi treatment augments MHC-I and MHC-II, but not PD-L1 expression on the cell surface upon IFNγ stimulation. **a** Viability assay of MMTV-Neu cells under guadecitabine (DMTi) treatment. All data are means ± SD. Dotted lines represent doses used in subsequent studies (0.5 and 0.05ug/mL). **b** Representative anti-5-mC dot blot of DNA isolated from MMTV-Neu cells under different concentrations (μg/ml; blot map in lower panel) of DMTi treatment. **c** Quantification of (**b**) based on integrated density of each dot. All data are means ± SD. **d** Fold change of GMFI of MHC-I, MHC-II and PD-L1 in MMTV-Neu cell line with guadecitabine treatment + IFNγ stimulation. GMFI ratio of diluent group was used as the baseline. *$p < 0.05$; **$p < 0.01$ (one-way ANOVA with Tukey's post-hoc test). All data are mean ± SEM. Each data point represents one independent experimental result. **e** Methylation-specific PCR for CpG-rich region of the H-2D$^q$ promoter in MMTV-neu cells. Genomic DNA isolated from cells 7 days after treatment with 0.5 μg/mL or 2 μg/mL guadecitabine were bisulfite-treated and analyzed by PCR using paired PCR primers specific for modified (methylated) and unmodified (unmethylated) cytosines. **f** Quantitative real-time PCR using three primer sets targeting mRNA expression of the H-2D$^q$ allele of MMTV-neu cells after 7 days after treatment with 0.05 μg/mL or 0.5 μg/mL guadecitabine or diluent control and with or without 24 h of stimulation with IFNγ (100 ng/mL)

dependent decrease in DNA methylation in the tumors from mice treated with guadecitabine (Fig. 3a). Since MHC-I, which presents antigen to cytotoxic CD8+ T cells, was most robustly altered with guadecitabine in vitro, we asked whether there were greater CD8+ infiltrates in guadecitabine-treated tumors using immunohistochemistry (IHC). Although there were no significant changes in total CD8+ T cell infiltrate, we observed a significant increase in the proportion of CD8+ T cells infiltrating into the tumor region compared to the stromal parenchyma ($p = 0.024$; Fig. 3b). This effect was confined to CD8+ T cells as total T cell infiltrate and CD4+ T cell infiltrate was not altered (Supplementary Figure 1). Flow cytometry analysis of MHC-I on EpCam+ tumor cells demonstrated an upregulation of MHC-I on tumor cells with DMTi, consistent with our findings in vitro. To substantiate this result, we utilized $Zr^{89}$-labeled anti-mouse CD8 antibody and positron emission tomography (PET) to visualize CD8+ T cell responses in vivo in tumor-bearing mice. Consistent with our observations by IHC analysis, we observed significantly increased ($p = 0.044$) CD8 accumulation in the central region in the tumors of DMTi treated mice (Fig. 3d, e). At 7 days following treatment initiation, we did not observe direct cytostatic or cytotoxic signal from guadecitabine as measured by Ki67 and TUNEL staining of tumors, although Ki67 trended toward decreased expression (Supplementary Figure 2a-b). It is possible that DMTi treatment could have an effect on regulatory immune function. However, no changes were observed in infiltrating myeloid (CD68+) or regulatory T cells (Foxp3+) in DMTi-treated tumors, although this does not eliminate the possibility of effects on functionality (Supplementary Figure 3). Nonetheless, we found that a single treatment cycle of guadecitabine induced complete responses in 4 of 7 mice treated, and stable disease in 2 of the remaining 3 mice, suggesting a prolonged immunologic effect on anti-tumor immunity mediated by guadecitabine treatment (Fig. 3f, g). It is worthy to note that 1 in 8 mice treated with control diluent spontaneously rejected its tumor, suggesting that this model is moderately immunogenic at baseline, likely due to the presence of the rat Neu xenoantigen[23]. Furthermore, mice treated with guadecitabine bearing smaller tumors (100–150 mm$^3$) were more likely to experience complete rejection. Finally, a subset of these mice (both control and guadecitabine-treated) were utilized for PET imaging, which includes micro-dosing of anti-CD8a antibody; in some studies treatment with anti-CD8a antibodies have been shown to potentiate effector function[24]. Nonetheless, these data suggest that targeting DNMT1 in breast tumors can upregulate MHC-I mediated antigen presentation and tip the balance at equilibrium to elicit a CD8+ T cell response which promotes tumor regression and anti-tumor immunity.

To confirm a T cell-mediated component to the activity of DMTi in vivo, a similar experiment was performed comparing growth of MMTV-neu cells in FVB/n or athymic nude mice, demonstrating that guadecitabine treatment had a superior effect in immunocompetent hosts ($p = 0.006$). Nonetheless, substantial innate anti-tumor activity was observed in the immunocompromised hosts with DMTi (Supplementary Figure 4).

**Activation of NFκB in tumors and cells after DMTi.** We next performed gene expression analysis of control and DMTi-treated MMTV-neu tumors using the NanoString mouse Pan-Cancer Immunology panel (771 gene transcripts) (Fig. 4a, Supplementary Data 1). Genes increasing in DMTi-treated tumors were associated with antigen presentation (*H2-D1*, *H2-K1*, *H2Q1*, *Tapbp*, *Tap1*, *Nlrc5* all MHC-I). Other genes upregulated in vivo after DMTi included genes associated with NFκB activity and interferon signaling. Many of these transcripts (e.g., *Irf7*, *Tlr3*, *Ifit1*, *Ifit3*, *Ifi44*, *Ifi35*, and *Tnssf10*) are signals of innate inflammatory

responses, and are consistent with previously described mechanisms of innate inflammation through demethylation-induced re-expression of endogenous retroviral (ERVs) sequences[14]. Re-expression of ERVs activates pattern-recognition receptors (PRRs) leading to induction of inflammation through NFκB[16,17]. Therefore, we asked whether DMTi treatment activated NFκB in breast cancer, and whether this was responsible for upregulation of MHC-I.

High doses (0.5 µg/mL) of guadecitabine clearly activated NFκB in cells, regardless of IFNγ treatment (Fig. 4b). However, blocking IκB/IKK with BMS-345541[25] did not significantly inhibit guadecitabine-induced basal MHC-I expression, while it strongly blocked IFNγ-induced MHC-I expression, regardless of DMTi (Fig. 4c). Similarly, T cell-recruiting Cxcr3 ligands (*Cxcl9/10/11*)[26] were strongly upregulated at the transcript level following DMTi, both basally and in response to IFNγ (Fig. 4d), similar to previous observations[26]. Basal upregulation of mRNA expression of Cxcr3 ligands following DMTi treatment was also largely independent of NFκB activation (Fig. 4e). Collectively, these results indicate that multiple mechanisms mediate MHC-I induction with DMTi, where demethylation of the MHC-I promoter may upregulate basal MHC-I levels, but also prime the promoter for NFκB induction (via IFNγ and ERV/PRR signaling) through epigenetic chromatin accessibility. Furthermore, although not directly tested in this study, the data support a multipronged model where DMTi-mediated upregulation of Cxcr3 chemokines enhance T cell recruitment, and MHC-I upregulation on tumor cells enhances recruited T cell activation.

**Epigenetic modulators enhance MHC-I expression in breast cancer patients.** To test whether inhibition of DNA methyltransferase activity can induce upregulation of MHC genes in human breast cancer patients, we explored gene expression data from longitudinally-collected biopsies of breast cancer patients treated with the DNA methyltransferase inhibitor 5′-azacytidine in combination with entinostat (HDAC inhibitor)[11]. In the five breast cancer patients on this multicenter phase II study of 5′-azacytidine and entinostat in women with advanced hormone-resistant or triple-negative breast cancer, where matched pre- and post-treatment biopsies were available (including a single patient with 2 month [Patient 5] and 6 month [Patient 5b] sequential post-treatment biopsies), there was a substantial and near-universal upregulation in MHC-I coupled with a more heterogeneous MHC-II response (Fig. 5a). These findings are consistent with the upregulation of antigen processing gene signatures reported in the initial analyses of the trial data[11]. Thus, DNMT1 inhibition may stimulate upregulation of MHC-I in a variety of breast cancers, and potentiate anti-tumor immunity. These effects could promote response to immunotherapies that leverage a restoration of adaptive equilibrium in the tumor microenvironment, such as PD-1-targeted therapy, in breast cancer. Furthermore, other molecularly-targeted inhibitors which modulate DNMT1, such as CDK4/6 inhibitors[27] which are already approved in breast cancer, may be viable approaches for synergistic combination with immunotherapies.

**DMTi potentiates response to anti-PD-1/L1 targeted therapy.** To test whether guadecitabine could effectively prime tumors for response to immunotherapy, we allowed MMTV-neu tumors in syngeneic hosts to reach slightly larger average tumor volumes (200–300 mm$^3$), prior to beginning guadecitabine therapy as before, administered daily for 3 days, followed by intraperitoneal injection of anti-PD-L1 antibody twice weekly for 2 weeks or IgG control. When guadecitabine treatment was initiated in larger initial tumor volumes, we observed substantially delayed tumor

growth, while in combination treated mice, marked disease stabilization and tumor regression were noted in some mice (Fig. 5b). Anti-PD-L1 therapy alone had little effect on tumor growth. Tumor size at the end of a 4-week period was significantly smaller in the combination arm than guadecitabine alone ($p = 0.04$; Fig. 5c). Surprisingly, these results were achieved with only a single 3-day course of guadecitabine priming therapy. Importantly, none of the mice in this experiment were used for PET imaging, thus eliminating anti-CD8 antibody treatment as a potentially confounding factor. To confirm activity in an additional breast cancer model, we treated orthotopically-established

polyoma V middle T murine mammary tumors in a similar fashion. Again, additive efficacy was observed with the combination of guadecitabine and anti-PD-L1 (Fig. 5d, e).

## Discussion

Herein, we explored the effects of DMTi to promote anti-tumor immunity in breast cancer. Among multiple breast cancer cell lines, spanning diverse molecular subtypes, we found that DMTi potently upregulated MHC-I expression, in some cases upregulated basal levels, but in nearly all cases potentiating type-II interferon responses. Molecular studies demonstrated that this

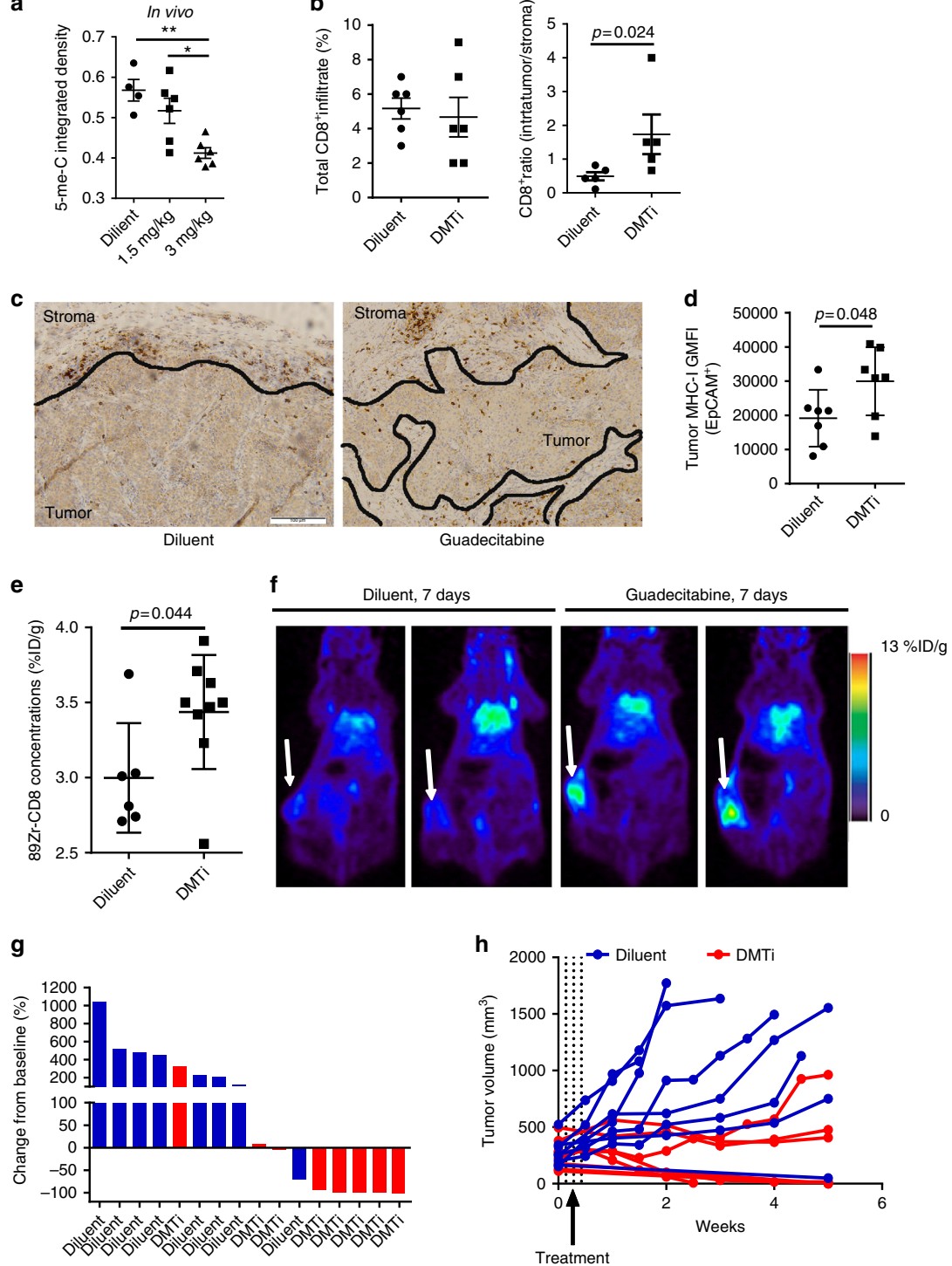

effect may have multiple contributing factors, including direct promoter demethylation of MHC-I genes, as well as stimulation of innate immunity, likely through previously described pattern recognition receptor activation and NFκB pathways[16,17]. However, our data support a multi-pronged effect where NFκB activation sensitizes the promoters to IFNγ activation, but basal levels of MHC-I (and possibly Cxcr3 chemokines) may be directly upregulated due to direct promoter demethylation.

In orthotopic animal models of breast cancer, DMTi with guadecitabine potentiated T cell recruitment, enhanced anti-tumor immunity, and exerted combinatorial activity with PD-L1-targeted immunotherapy. Given these results, combining DMTi with anti-PD-1/L1 targeted immunotherapy may be a viable approach to converting immunotherapy-refractory tumors, such as luminal-like breast cancers, to immunotherapy-responsive tumors, and should be tested in future clinical trials.

## Methods

**Cells and treatment.** Human breast cancer cell lines MCF7, BT474, BT549, and HCC1395 were obtained from ATCC. MCF7 and BT474 were grown in DMEM (Gibco) supplemented with 10% fetal bovine serum (FBS) (Denville). BT549 and HCC1395 were grown in RPMI (Gibco) supplemented with 10% FBS. The murine mammary carcinoma cell line, MMTV-Neu, was originally isolated from a primary mammary tumor (transgenic FVB/N mice) and cultured in DMEM/F12 (Gibco) supplemented with 10% FBS, 20 ng/ml EGF (Gibco), 0.5 μg/ml Hydrocortisone (Santa Cruz), and 10 μg/ml Insulin (Gibco). MCF7, BT474, BT549, HCC1395, and MMTV-Neu cells were pre-treated with diluent (Astex Pharmaceuticals), 0.05 μg/ml guadecitabine (Astex Pharmaceuticals) or 0.5 μg/ml guadecitabine for 3 days and stimulated with 100 ng/ml IFNγ (Gibco) for an additional 3 days with or without the presence of diluent control or guadecitabine. All cell lines were routinely tested for mycoplasm contamination.

**Flow cytometry.** For in vitro analysis, cells were washed with phosphate-buffered saline and dissociated from the plate with accutase (Gibco) for 5–10 min at 37 °C to generate single cell suspensions. For in vivo studies, tumors were excised post-mortem, mechanically digested using gentle MACS tubes (Miltenyi) and enzymatically digested using a mixture of 0.5 mg/ml collagenase type III (Sigma-Aldrich), 0.01 mg/mL dispase, and 0.125 mg/ml DNAase (Sigma-Aldrich) with antibiotic, 30-min at 37 °C. Tissue dissociates were passed through a 70-μm filter to collect a single cell suspension. Single cell suspensions were washed once in flow staining buffer and incubated with respective flow antibodies at 4 °C for 30 min in the dark. DAPI was used to discriminate viable and dead cells. Tumor cells were gated on EpCAM-positive cells. Flow cytometry was performed using the following antibodies: HLA-A,B,C/Alexa Fluor488 (Biolegend, clone W6/32, 1:200), HLA-DR/PE-Cy7 (Biolegend, clone L243, 1:200), mouse MHC-I (H-2Kd/H-2Dd)/PE (Biolegend, clone 34-1-2S, 1:200), mouse MHC-II (I-A/I-E)/Alexa Fluor488 (Biolegend, clone M5/114.15.2, 1:200), mouse CD274(PD-L1)/APC (Biolegend, clone 10F.9G2, 1:200), and mouse EpCAM/PE-Cy7 (Biolegend, clone G8.8, 1:350). Samples were analyzed on an Attune NxT system (Life Technologies).

**Viability assays.** MMTV-Neu cells were plated at a density of $10^3$ cells per well in a 96-well plate and treated with a 2-fold dilution series of guadecitabine for either 3 or 7 days. Viability was ascertained with sulfarhodamine B (SRB) (ACROS). In brief, cells were fixed with 10% trichloroacetic acid (TCA) at 4 °C for 30 min then stained with 0.4% SRB at room temperature for 10 min. Plates were air-dried, then

SRB re-solubilized with 10 mM Tris-HCl, pH 7.5 and quantified by absorbance (490 nm) and normalized to control (DMSO treated).

**Dot blot for 5-methylcytosine.** Genomic DNA was isolated from tumors or tumor cell lines (Maxwell, Promega), denatured at 99 °C for 5 min, and 1 μg DNA was spotted on positively charged nylon membranes and air-dried. The membrane was UV cross-linked and blocked (5% non-fat dry milk, tris-buffered saline, 0.1% Tween-20) for 1 h at room temperature. The membrane was incubated overnight at 4 °C with anti- 5-mC antibody (EpigenTek, A-1014, 1:500). Following incubation with horseradish peroxidase-conjugated anti-mouse secondary antibodies (Santa Cruz, 1:5000) for 1 h at room temperature, proteins were visualized using an enhanced chemiluminescence detection system (Thermo). The 5-mC signal was quantified by determining the integrated density of each dot using ImageJ.

**DNA methylation-specific PCR.** MMTV-Neu cells were plated at $3 \times 10^5$ cells per well of a 6 well dish. Cells were treated on days 1 and 4 with 0.5 μg/ml or 2.0 μg/ml guadecitabine or diluent. On day 7, plates containing cells were placed on ice, washed once with 1X D-PBS, harvested by scraping in 1X D-PBS, and DNA isolated as per manufacturer's specifications (Promega Maxwell 16 DNA Purification Kit). Bisulfite conversion and cleanup was performed on 1 μg of genomic DNA as per manufacturer's specifications (EpiTech Bisulfite Kit, Qiagen). Methylation specific PCR was performed using 25 ng of converted DNA in 1X PCR buffer II (Applied Biosystems), 2 mM MgCl2, 0.2 mM each dNTP, 0.6 μM forward primer, 0.6 μM reverse primer, and 1.0 UAmpli-taq Gold (Applied Biosystems). Cycling parameters were 95 °C for 10 min, 94 °C for 30 s, followed by 40 cycles of 58 °C for 30 s, 70 °C for 30 s, and 72 °C for 5 min. The PCR product was resolved on a 2% agarose gel. Methylated and unmethylated primer sequences are shown in Supplementary Table 1.

**Quantitative real-time reverse transcriptase PCR.** MMTV-Neu cells were treated as described above. Plates containing cells were placed on ice, washed once with 1XD-PBS (Life Technologies), harvested by scraping in chilled 1-Thioglycerol/Homogenization solution, and RNA isolated as per manufacturer's specifications (Promega Maxwell 16 LEV simply RNA Tissue Kit). cDNA was generated from 1 μg of total RNA (Bioline, SensiFAST™ cDNA Synthesis Kit) as per manufacturer's specifications. cDNA was diluted 1:5 and 5 μl (50 ng equivalents of RNA) was used as input for analysis in a reaction using 1X SsoAdvanced Universal SYBR Green Supermix (BioRad) and 5 μm each primer. Cycling parameters were 95 °C for 3 min, followed by 40 cycles of 95 °C for 15 s and 58 °C for 30 s. Primer sequences used for qRT-PCR are listed in Supplementary Table 2.

**Mouse studies.** Mouse procedures and studies were approved by the Vanderbilt Division of Animal Care and the Institutional Animal Care and Use Committee. Established MMTV-Neu mammary tumor cells or primary MMTV-polyoma V middle-T mammary tumor cells ($1 \times 10^6$) were orthotopically injected into the 4th mammary fat pads of FVB/n mice (or athymic nu/nu mice, for MMTV-Neu). Following the establishment of tumors (~100–200 mm³), the mice were treated with diluent or guadecitabine (3 mg/kg, I.P. injection, 3 continuous days), or in combination with isotype IgG (BioXcell, clone LTF-2, 100 μg intraperitoneal, on days 3, 7, 10, 14) or α-PD-L1 (BioXcell, clone 10F. 9G2, 100 μg intraperitoneal, on days 3, 7, 10, 14). For T cell infiltration analysis, mice were euthanized on Day 7 after the initiation of treatment and tumor samples were collected for IHC. Six to eight (6–8) mice were used for each treatment arm for these studies. For tumor growth analysis, tumor was measured 2–3 times weekly with calipers and volume was calculated in mm³ using the formula (length x width x width/2). Mice were humanely euthanized when the tumor volume reached 2 cm³ or 5 weeks initiation of treatment (4–5 weeks for the combination study). At least five mice were used for each treatment arm for tumor growth studies.

**Fig. 3** DMTi treatment increased tumor-infiltrating CD8+ T cells and promotes tumor regression in vivo. **a** Representative anti-5-mC dot blot of DNA isolated from MMTV-Neu tumors under in vivo administration of different concentrations of guadecitabine (DMTi) and corresponding quantification based on integrated density. * $p < 0.05$ (one-way ANOVA with Dunnett's post-hoc test). All data are means ± SEM. Each dot represents one experimental result. **b** Evaluation of CD8+ infiltration in MMTV-Neu tumors by IHC under in vivo administration of diluent vs DMTi. The result was illustrated as percentage of CD8+ over the whole tumor section (left) or the ratio of CD8+ infiltration between intra-tumor and stroma compartments (right). P values were calculated using an unpaired t-test. All data are represented as mean ± SEM. Each dot represents data collected from one experimental mouse. **c** Representative CD8 + IHC of MMTV-Neu tumors after in vivo administration of diluent vs DMTi. The black line outlines margins between tumor and stroma compartments. Scale bar represents 100 μm. **d** GMFI of MHC-I in the EpCAM+ gated population of dissociated MMTV-Neu tumor samples following in vivo administration of guadecitabine or diluent control. P-values were calculated using an unpaired t-test. All data are means + SEM. Each dot represents one dissociated tumor sample from one experiment mouse. **e** Quantification of [$^{89}$Zr]CD8 radioactivity by ImmunoPET in mice orthotopically injected with MMTV-Neu cells under in vivo administration of Diluent vs DMTi. The p-value was calculated using unpaired t test. All data are means + SEM. Each dot represents one experimental mouse. **f** Representative ImmunoPET images of mice orthotopically injected with MMTV-Neu cells and treated with guadecitabine or diluent control (7 days post-therapy initiation). **g** Waterfall plot of MMTV-Neu tumor volume following treatment with guadecitabine or diluent control. The result was illustrated as the percentage of tumor volume change from baseline. **h** Tumor growth curve of MMTV-Neu tumors following in vivo administration of diluent vs DMTi on days 1–3

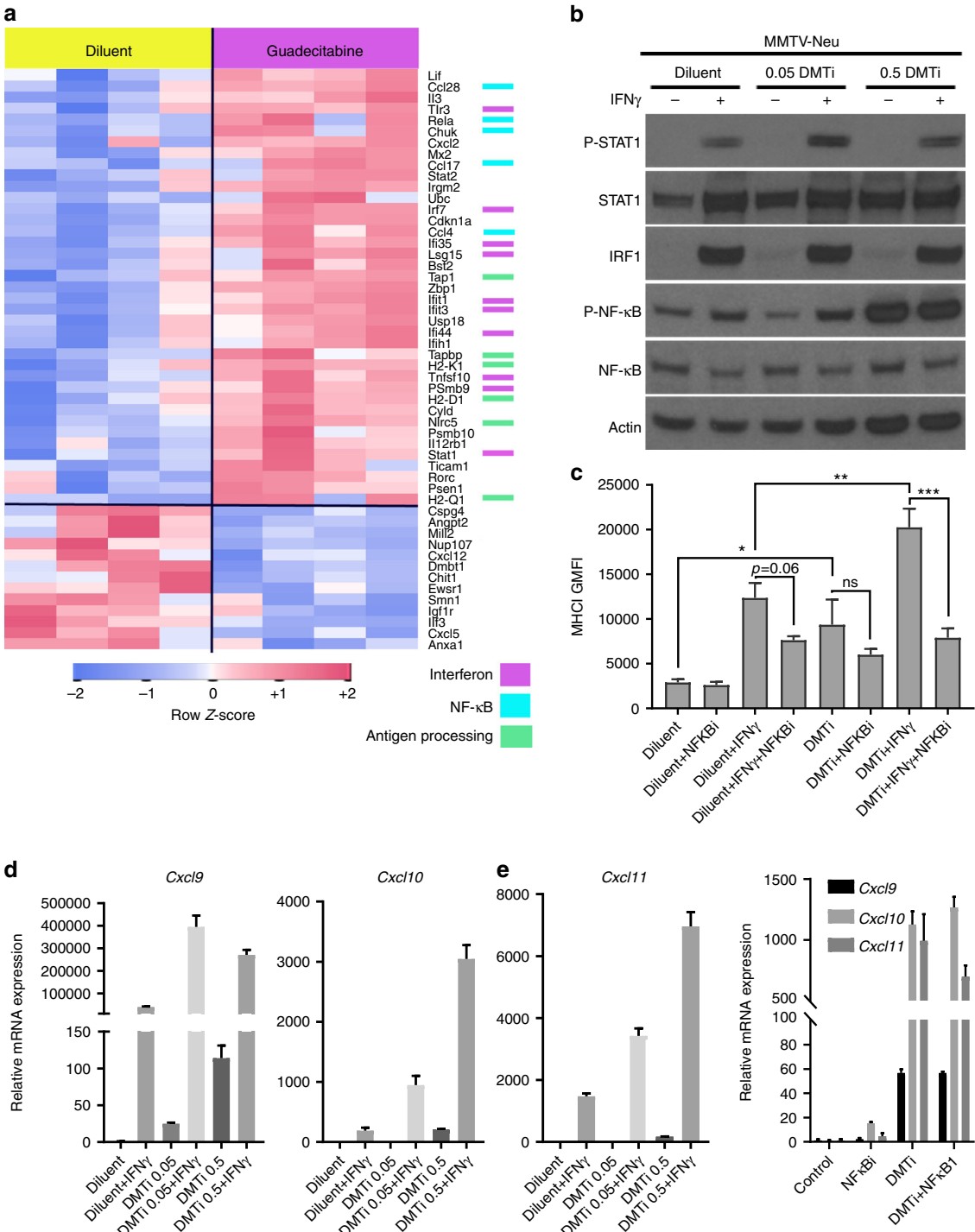

**Fig. 4** Guadecitabine enhances IFNγ-mediated MHC-I expression and Cxcr3 ligands. **a** MMTV-neu tumors were resected 7 days after a 3-day in vivo treatment course with guadecitabine or diluent control. Tumor RNA was extracted and utilized for NanoString gene expression analysis using the PanCancer Immune Pathways codeset (>700 immune-related genes). Genes significantly altered (nominal p-value < 0.05) between the treatment groups are shown with a row-standardized z-scores in heatmap form. Association of altered genes with key pathways are color coded on the right. **b** MMTV-neu cells were cultured for 7 days with the indicated doses of guadecitabine, followed by 24 h treatment with IFNγ and evaluation by western blot analysis. **c** MMTV-neu cells were treated as in **b** but co-treated with or without BMS-345541 (during last 24 h, in tandem with IFNγ) and assessed by flow cytometry for MHC-I. Data represent the mean of four experiment replicates ± SEM. *p < 0.05; **p < 0.01; ***p < 0.001; ns nonsignificant. **d** qRT-PCR for Cxcr3 ligands in MMTV-neu cells after 7 days of guadecitabine treatment at the indicated doses and 24 h treatment with IFNγ. Data represent the mean of 3 experiment replicates ± SEM. **e** Relative mRNA expression of Cxcr3 ligands in MMTV-neu cells after 7 days of treatment with guadecitabine (0.5 μg/mL) or diluent control, and 24 h exposure to 10 μM BMS-345541. Data represent the mean of 3 experiment replicates ± SEM

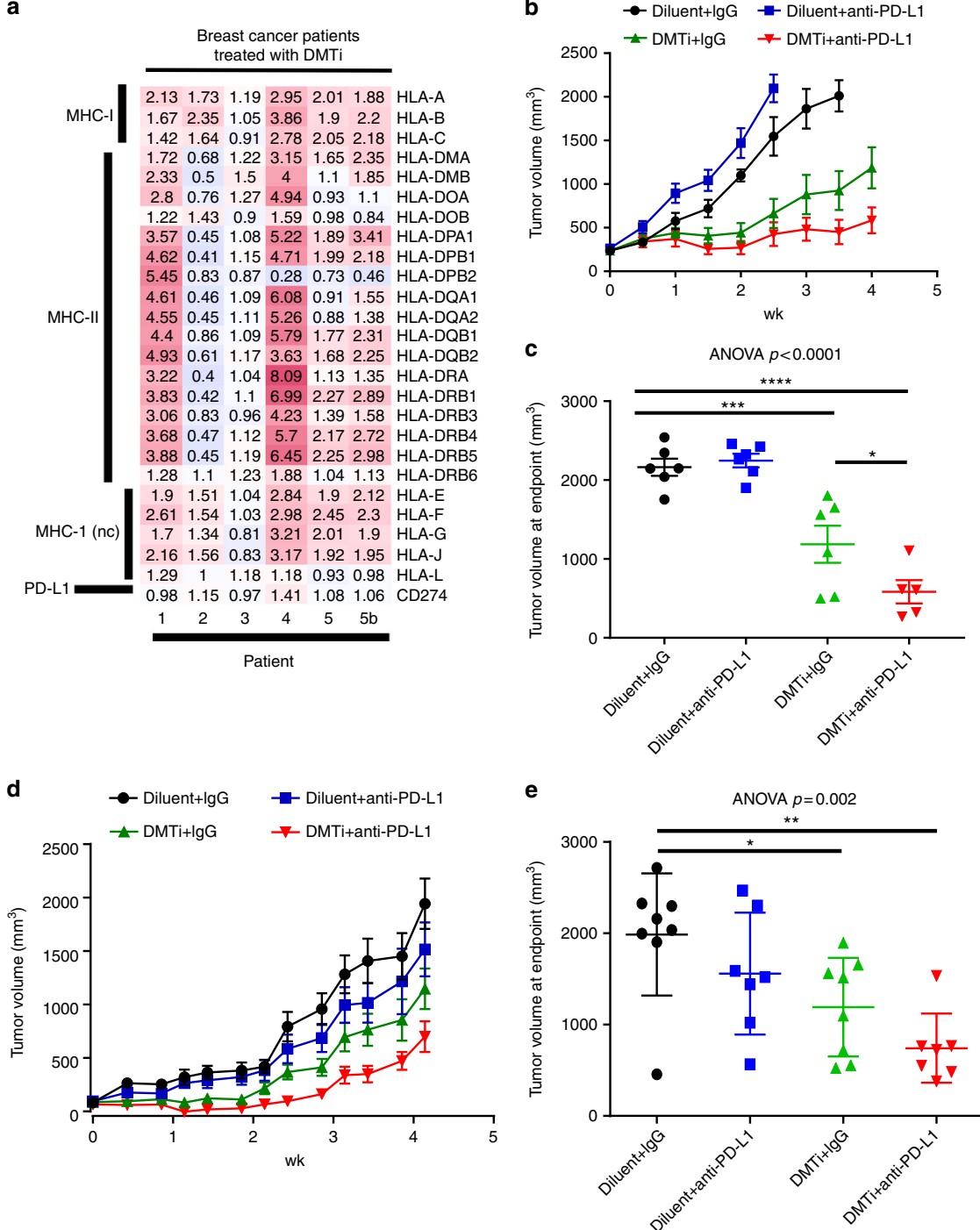

**Fig. 5** Epigenetic treatment augments MHC-I and MHC-II in breast cancer patient samples, and potentiates response to anti-PD-L1 therapy in mice. **a** The heatmap illustrates post/pre-treatment mRNA expression ratio of MHC-I, MHC-II, and PD-L1 in 5 pairs of triple negative breast cancer patient samples. Each pair includes "Pre" (before 5'-azacitidine, AZA, and entinostat treatment) and "post" (ratio of 8wks:baseline in patients 1–5 or ratio of 6mos: baseline [patient 5b] after AZA and entinostat treatment). MHC-I nc: non classical MHC-I **b** Tumor growth curves from MMTV-Neu tumor-bearing FVB/n mice treated with guadecitabine daily for 3 days, followed by twice weekly anti-PD-L1 for 2 weeks (or appropriate controls). Mice were sacrificed when tumor volume measurements first exceeded 2 cm$^3$, or at 4 weeks whichever occurred first. **c** Final tumor volumes at the humane endpoint by treatment group from (**b**). Differences in final tumor volume (mean ± SEM) were compared by ANOVA followed by Tukey's post-hoc test to compare relevant groups. *$p < 0.05$; ***$p < 0.001$; ****$p < 0.0001$ for individual comparisons. **d** Tumor growth curves from MMTV-PyVmT tumor-bearing FVB/n mice treated with guadecitabine daily for 3 days, followed by twice weekly anti-PD-L1 for 2 weeks (or appropriate controls). Mice were sacrificed when tumor volume measurements first exceeded 2 cm$^3$, or at 4 weeks whichever occurred first. **e** Final tumor volumes at the humane endpoint by treatment group from (**d**). Differences in final tumor volume (mean ± SEM) were compared by ANOVA followed by Tukey's post-hoc test to compare relevant groups. *$p < 0.05$; **$p < 0.01$ for individual comparisons

**NanoString analysis**. Gene expression analysis on MMTV-neu tumors following a single course of guadecitabine or diluent control were performed using the nanoString Pan-cancer immunology panel according to the manufacturers' standard protocol and according to previous analyses[3]. Briefly, single cross sections of residual tumors following 14–17 days of treatment were used for RNA preparation and 50 ng of total RNA > 300nt was used for input into nCounter hybridizations. Data were normalized according to positive and negative spike-in controls, then endogenous housekeeper controls, and transcript counts were log transformed for statistical analyses.

**Immunohistochemistry**. For all staining, tissues were routinely processed, and antigen retrieval was performed using citrate buffer (pH = 6) or Tris EDTA buffer (pH = 9), using the Decloaking Chamber from Biocare. After exhaustion of endogenous peroxidase with hydrogen peroxide, slides were blocked with Protein Block Solution (Dako) for 10 min at room temperature and incubated with primary antibody overnight at 4 °C. Primary antibodies were as follows: CD3 (Abcam, Ab16669, 1:800), CD4 (eBioscience, 14-9766-80, 1:1000), and CD8 (eBioscience, 14-0195-82, 1:100), CD68 (BIO-RAD, catalog# MCA1957; dilution 1:100), Foxp3 (eBioscience, 13-5773-82 1:50), Ki67 (Biocare, Catalog# CRM325B; dilution 1:100) and TUNEL staining was performed using the ApopTag kit (Millipore, Catalog# S7100) according to the manufacturer's recommended protocol. Visualization was performed using Envision (Dako) and DAB (Dako). Samples were de-identified, and the percentage of marker-positive cells were evaluated by a breast cancer pathologist.

**Preparation of [⁸⁹Zr]Zr-DFO-CD8a**. 1 mL of CD8a antibody (BioXcell, clone 53-6.72) solution (5.95 mg/mL, 186 nmol) was adjusted to a pH of 9.0 by addition of 0.1 M Na₂CO₃ (300 µL). Three fold molar excess of DFO-Bz-NCS (28 µL of 20 mM DFO in DMSO) was added to the CD8a solution and the reaction mixture was incubated with a gently shaking for 1 h at 37 °C in a heated water bath. [⁸⁹Zr]Zr oxalate in 1 M oxalic acid (10.05 mCi, 371.85 MBq, 200 µL) was neutralized (pH = 7.0–7.5) by slow addition of 1 M Na₂CO₃ solution (~190 µL). 100 µL of the CD8a-DFO-Bz-NCS conjugate solution (450 µg of CD8a) was added to a mixature of 300 µL of 0.5 M HEPES (pH = 7.0) buffer and 120 µL of gentisic acid buffer. The [⁸⁹Zr]Zr-oxalate solution was then added and the reaction mixture was incubated for 1 h at room temperature with occasional gentle shaking. The reaction mixture was transferred onto a PD10 column previously equilibrated with 20 mL of water (Traceselect) and then 4 mL of gentisic acid buffer. A volume of 8 mL of the gentisic acid buffer was pipetted onto the column and the [⁸⁹Zr]Zr -DFO-CD8a was collected in 0.5 mL fractions. Radiochemical purity (RCY) of each fraction was determined by iTLC. (iTLC (SG) was performed using separate two solvent systems (i) 20 mM of citric acid solution (ii) ethanol:ammonium hydroxide:water (2:0.5:5). Radioactivity was counted in Bioscan AR2000 Radio-TLC Imaging scanner.) The fractions were combined and passed through a 0.45 µM filter into a sterile vial (pH 6.5–7.0, RCY > 94%).

**ImmunoPET studies**. MMTV-Neu cells ($1 \times 10^6$) were orthotopically injected into the #4 mammary fat pad of *FVB/n* mice (*n* = 15). Following the establishment of tumors (~100–200 mm³) as measured by calipers, the mice were treated with diluent or guadecitabine (3 mg/kg, I.P. injection, 3 consecutive days). Seven days post-treatment, the mice were injected intraperitoneally with ~ 8 MBq of [⁸⁹Zr] CD8 and imaged 24 h later in a microPET Focus 220 (Simens, Knoxville TN) for 30 min. Three-dimensional donut-shaped regions-of-interest (ROIs) were drawn around tumors using Amide (www.sourgeforge.net). The radiotracer concentration within the ROIs were normalized to the total injected dose and expressed as percent-injected dose/gram of tissue (%ID/g). At least six mice were used for each treatment arm (diluent control *n* = 6; guadecitabine *n* = 9).

**Epigenetic therapy-treated patients and microarray data**. Microarray data from patient biopsies from a Phase II Study of Azacitidine and Entinostat (SNDX-275) in Patients With Advanced Breast Cancer NCT01349959 led by Dr. Vered Stearns were kindly provided by Johns Hopkins University. All patients were administered informed consent and received 40 mg/m² 5-azacitidine subcutaneously on days 1–5 and 8–10 and 7 mg oral entinostat on days 3 and 10. Courses were repeated every 28 days in the absence of disease progression or unacceptable toxicity. RNA was isolated from pre- (baseline) and post-treatment (8 weeks (*n* = 5) and 6 months (*n* =1)) biopsies and analyzed with the Agilent 44 K Expression Array. Data were normalized in R by using limma. 'loess' and 'aquantile' packages were used for within- and between-array normalization. Probes were collapsed to gene names by median, such that each gene has only one value. Initial analyses of these data have been previously described[11].

**Statistical analysis**. Statistical analyses were performed as indicated using *R* or GraphPad Prism (GraphPad Software). For two-group analyses, Student's *T* test was used to compare means, except in cases where the variance was significantly different between the groups, in which cases a non-parametric equivalent was utilized (Mann–Whitney U). In comparisons among multiple groups, a one-way ANOVA was used with post-hoc correction for between between-group comparisons as indicated.

**Data availability**. All relevant data are available upon request from the authors. Normalized count data for NanoString nCounter analysis are available in Supplementary Data 1.

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

## Acknowledgements

Funding for this work was provided by The V Foundation for Cancer Research (V2016-012, JMB), The China Scholarship Council (201406205050, N.L.), the NIH/NCI (R00CA181491, J.M.B.), Susan G. Komen for the Cure Foundation CCR14299052 (J.M.B.), the Breast Cancer Specialized Program of Research Excellence (SPORE) P50 CA098131, the Vanderbilt-Ingram Cancer Center Support Grant P30 CA68485, The Kleburg Foundation, Trans-Institutional Programs (TIPS) award to the Center for Molecular Probes, and the Vanderbilt University Office of Research support of the Radiochemistry Core Resource.

## Author contributions

N.L. and J.M.B. co-wrote the manuscript. J.M.B. was responsible for overseeing the work. N.L. designed and performed the experimental analyses with additional contributions from S.R.O. and M.N. P.I.G.E., V.S., and M.E.S. performed immunohistological analyses and pathology-related scoring of samples. C.A.Z. and H.L. provided data from breast cancer patients. M.N.T., H.C.M., F.L., and M.L.N. optimized and performed ImmunoPET imaging studies. All authors provided expert feedback and editing of the final manuscript.

## Additional information

**Competing interests:** The authors declare no competing financial interests.

