## [Peer Review File · Nature Communications]

Reviewers' comments:

Reviewer #1 (Remarks to the Author):

In this manuscript Dr. Na Luo and coll, report on the activity of the DNA methyltransferase inhibitor guadecitabine in upregulating the expression of HLA antigens in breast cancer cells, and in favouring the influx of T cells in breast cancer through an IFN-regulated mechanism. Based on their findings the Authors conclude that the use of DNA methyltransferase inhibitors could represent a useful approach to potentiate the anti-tumor T cell responses to breast cancer cells and the efficacy of checkpoint inhibitors therapy in an otherwise immune-refractory disease.

The findings reported in this manuscript are not novel as the immunomodulatory potential of DNA methyltransferase inhibitors has been extensively investigated in different tumor types and extensive mechanistic insights have been published to support the use of these agents to improve tumor recognition by antigen specific CTL. Therefore, the field will get limited influence by the results present in this manuscript. The Authors' would also greatly benefit from reporting the extensive available literature that highlights the phenotypic and functional immunomodulatory activity of DNA methyltransferase inhibitors (e.g., HLA antigens, costimulatory/antigen pre-processing molecules, tumor associated antigens, IFN-pathway, etc, etc) and the concept that they could represent useful agents to improve the efficacy of cancer immunotherapy (just as possible examples please see the following review articles: Sigalotti et al, *Pharmacol Ther.* 2014; Wolff et al, *Cell Communication and Signalling* 2017).

Reviewer #2 (Remarks to the Author):

This interesting manuscript from Luo, et al presents data measuring the immunomodulatory effect of DNMTi in both human and mouse breast cancer cell line models. They found that treatment with guadecitabine upregulated MHC-I expression in response to interferon- γ stimulation. Further, MHC-I and in some cases, MHC-II genes but not PD-L1 were upregulated in breast cancer patients treated with hypomethylating agents. In a murine tumor model of breast cancer, guadecitabine upregulated MHC-I+ tumor cells and induced recruitment of CD8+ T cells from the tumor parenchyma to the intra-tumor microenvironment. Finally, they showed that treatment with guadecitabine promoted the efficacy of anti-PD-L1 therapy, suggesting that combining DNMTi with anti-PD-1/L1 targeted immunotherapy may be a viable approach for clinical development. Although this study holds some levels of interests, it is not well developed into scientifically sound contribution in its current form.

Specific comments:

1. Descriptive nature of the study limits potential impact. For example, at least some mechanistic experiments are needed to understand how guadecitabine modulate MHC-I expression (and in some cases, MHC-II) in response to interferon- γ stimulation.

2. In a murine tumor model of breast cancer, guadecitabine upregulated MHC-I+ tumor cells and induced recruitment of CD8+ T cells into the tumor. However, it is unclear if antitumor effect of guadecitabine treatment is really dependent on antitumor CD8 T cell immunity. Did guadecitabine alter the phenotype, survival and functionality of these tumor-infiltrating CTLs? Did guadecitabine enhance the tumor cell recognition by CTL because of upregulated MHC-I? How did guadecitabine promote CTL recruitment? The authors should experimentally address these questions, as they are important points that constitute the major novelty of the report.

3. The study skims over the possibility that the DNMTi might have direct effects on antitumor T-cells (see *Nature.* 2015;527:249-53). It would be also important to determine if DNMTi directly affects the other key immune cell components (such as Tregs and myeloid cells).

4. The paper may be strengthened by comparing the current tumor model to at least one other mouse model of breast cancer.

Response to Reviewers' comments:

Reviewer #1:

1. The findings reported in this manuscript are not novel as the immunomodulatory potential of DNA methyltransferase inhibitors has been extensively investigated in different tumor types and extensive mechanistic insights have been published to support the use of these agents to improve tumor recognition by antigen specific CTL. Therefore, the field will get limited influence by the results present in this manuscript.

We appreciate these concerns. Nonetheless, we would argue that the data present a novel compilation of the established findings of others in a tumor type for which both demonstration and translation of these mechanisms has largely been underexplored/not performed. The immune-tumor microenvironment and cell signaling pathways activated and engaged (or disengaged) are noticeably different between tumor types. This concept is often ignored or underexplored in tumor immunology, but has been the subject of decades of research in cancer biology. Thus, the demonstration of these findings in breast cancer hold particular relevance to the translation to clinical studies in a tumor type which could stand to benefit greatly to combination therapy as single-agent immunotherapy has demonstrated limited efficacy. In addition to this point, our revised manuscript now includes the novel demonstration that guadecitabine treatment directly demethylates the MHC-I promoter, likely contributing to its upregulation.

2. The Authors' would also greatly benefit from reporting the extensive available literature that highlights the phenotypic and functional immunomodulatory activity of DNA methyltransferase inhibitors (e.g., HLA antigens, costimulatory/antigen processing molecules, tumor associated antigens, IFN-pathway, etc) and the concept that they could represent useful agents to improve the efficacy of cancer immunotherapy (just as possible examples please see the following review articles: Sigalotti et al, Pharmacol Ther. 2014; Wolff et al, Cell Communication and Signalling2017).

We thank the reviewer for identifying these reviews and have key references included in the reviews as well as the reviews themselves, to the manuscript (ref 12-17). We are happy to include additional references if needed.

Reviewer #2:

We sincerely thank the reviewer for their time and critical review of our manuscript and appreciate the advice on enhancing the work through additional mechanistic detail. In the 3 month advisement by the editor, we have completed the following additions to the work, which we hope addresses at least the majority of the Reviewer's concerns.

1. Descriptive nature of the study limits potential impact. For example, at least some mechanistic experiments are needed to understand how guadecitabine modulate MHC-I expression (and in some cases, MHC-II) in response to interferon- γ stimulation.

We now present additional data in Figure 4 (new figure) analyzing gene expression from MMTV-neu DMTi-treated tumors (Figure 4a). These findings led us to determine whether pattern recognition receptors (PRR), known to be induced upon endogenous retroviral sequence (ERVs) re-expression following DMTis, and subsequent NF κ B activation were responsible for upregulation of MHC-I (Figure 4b-c).

These studies identified a direct effect of NFκB in induction of MHC-I following interferon-γ stimulation, but also showed that demethylation of the MHC-I (H-2D) promoter is likely responsible for MHC-I basal upregulation in response to DMTi (Figure 2e-f, shown below). We hope these additional insights add significant mechanistic insight and novelty to the work.

2. In a murine tumor model of breast cancer, guadecitabine upregulated MHC-I+ tumor cells and induced recruitment of CD8+ T cells into the tumor. However, it is unclear if antitumor effect of guadecitabine treatment is really dependent on antitumor CD8 T cell immunity.

In order to address this concern, we now demonstrate that the same tumor model, grown in athymic nude mice, has reduced responses to guadecitabine ($p=0.006$), new Supplementary Figure 4, below, suggesting that activity is, in part, T cell mediated.

a) Did guadecitabine alter the phenotype, survival and functionality of these tumor-infiltrating CTLs?

This is an important, but broad, question that is experimentally difficult to address, as the total T cells in the tumor microenvironment was less significantly altered as compared to the infiltrating ones. It would be experimentally challenging to dissect the associated stroma to evaluate survival and functionality. However, enhanced multiplexed IHC/IF, or new strategies to isolate T cells from specific compartments within the tumor may help drive an experimental model to answer this question in future studies. Although an excellent question that could be considered in the scope of the presented work, we are unable to perform these experiments in the allotted time without ruling out a direct effect of DMTi on immune cells (rather than an indirect effect being mediated through the tumor). This is certainly an area of ongoing study however.

b) Did guadecitabine enhance the tumor cell recognition by CTL because of upregulated MHC-I?

This is an extremely complicated experiment that we are, unfortunately, unable to perform. Certainly, approaching this from a loss-of-function perspective by eliminating MHC-I (CrispR/shRNA, for example targeting B2m) will eliminate CD8+ T-cell mediated immunity, thus providing falsely-positive results, and not directly answering the question (i.e. whether expression of MHC-I can modulate anti-tumor immunity in a dose-dependent manner). On the other hand, without doing extensive experimentation to derive the antigen(s) (and associated MHC-I allele) responsible for the effect, we would need to directly modulate the expression of H2-D/K/L independently by overexpression or induction. Nlr5 (a putative 'master regulator' of class I) overexpression could be used, but it is arguable as to the specificity of Nlr5 to only class I genes.

Although it is easy to conceive that upregulation of MHC-I would probabilistically increase TCR-MHC interactions, thereby promoting antitumor immunity, it is also possible that class I expression level is not a rate limiting step. We have added these points to the discussion, but unfortunately do not have the means to answer the identified question. We have made every attempt to present this caveat in the revised manuscript.

c) How did guadecitabine promote CTL recruitment?

In Figure 4d, below, we now demonstrate that DMTi enhances both the basal and interferon- γ -induced expression of Cxcl9, 10, and 11. These chemokines are known to bind to Cxcr3+ T cells, licensing them to leave the tumor draining lymph node and migrate to the site of inflammation. Thus, this is a potential mechanism of CTL recruitment.

- The study skims over the possibility that the DMTi might have direct effects on antitumor T-cells (see Nature. 2015;527:249-53).

To address these, we confirm Cxcl9 and Cxcl10 tumor cell upregulation with guadecitabine (shown above), which were the Th1-targeted chemokines identified by EZH2 inhibition in the cited manuscript (these chemokines were induced in the tumor cells themselves, which was the premise of that publication).

- It would be also important to determine if DMTi directly affects the other key immune cell components (such as Tregs and myeloid cells).

We have quantified infiltration of both myeloid (Cd68+) and regulatory T (Foxp3+) cells in the tumor microenvironment, with and without DMTi (new Supplemental Figure 3, below). We observed a slight increase in Foxp3+ T cells (% of total TILs only) and a modest decrease in myeloid cell infiltration (total). Neither was statistically significant at our sample size. However, this does not rule out possibility for functional changes, but a deeper exploration of this hypothesis would seem to be out of the scope of the present work.

Supplemental Figure 3

5. The paper may be strengthened by comparing the current tumor model to at least one other mouse model of breast cancer.

We now present confirmatory data in the PyVmt breast cancer model, which demonstrates similar results to the MMTV-neu model in Figure 5d-e, below.

REVIEWERS' COMMENTS:

Reviewer #2 (Remarks to the Author):

The MS has been greatly improved by adding more experiments. I have no further comments.